Adaptive Metropolis-coupled MCMC for BEAST 2

Müller Nicola F. 1 2 3 nicola.felix.mueller@gmail.com
http://orcid.org/0000-0001-6765-3813 Bouckaert Remco R. 4 5 r.bouckaert@auckland.ac.nz
1 Department of Biosystems Science and Engineering, ETH Zürich , Basel , Switzerland
2 Swiss Institute of Bioinformatics , Lausanne , Switzerland
3 Fred Hutchinson Cancer Research Center , Seattle, Washington , Switzerland
4 School of Computer Science, University of Auckland , Auckland , New Zealand
5 Max Planck Institute for the Science of Human History , Jena , Germany
Castro-Nallar Eduardo
Electronic publication date: 2020 Sep 16
Publication date: 2020
Volume: 8
Electronic Location ID: e9473
Received 2020 Jan 15; Accepted 2020 Jun 12
Copyright: © 2020 Müller and Bouckaert
Copyright year: 2020
Copyright holder: Müller and Bouckaert
License: This is an open access article distributed under the terms of the Creative Commons Attribution License, which permits unrestricted use, distribution, reproduction and adaptation in any medium and for any purpose provided that it is properly attributed. For attribution, the original author(s), title, publication source (PeerJ) and either DOI or URL of the article must be cited.
License URL: https://creativecommons.org/licenses/by/4.0/

Keywords: Bayesian, Phylogenetics, Phylodynamics, Coalescent, Parallel tempering

Funding: Swiss National Science foundation (SNF) CR32I3_166258 Royal Society of New Zealand 18-UOA-096 Nicola F. Müller is funded by the Swiss National Science foundation (SNF; grant number CR32I3_166258). Remco R. Bouckaert is supported by the Marsden grant 18-UOA-096 from the Royal Society of New Zealand. The funders had no role in study design, data collection and analysis, decision to publish, or preparation of the manuscript.

==============================
With ever more complex models used to study evolutionary patterns, approaches that facilitate efficient inference under such models are needed. Metropolis-coupled Markov chain Monte Carlo (MCMC) has long been used to speed up phylogenetic analyses and to make use of multi-core CPUs. Metropolis-coupled MCMC essentially runs multiple MCMC chains in parallel. All chains are heated except for one cold chain that explores the posterior probability space like a regular MCMC chain. This heating allows chains to make bigger jumps in phylogenetic state space. The heated chains can then be used to propose new states for other chains, including the cold chain. One of the practical challenges using this approach, is to find optimal temperatures of the heated chains to efficiently explore state spaces. We here provide an adaptive Metropolis-coupled MCMC scheme to Bayesian phylogenetics, where the temperature difference between heated chains is automatically tuned to achieve a target acceptance probability of states being exchanged between individual chains. We first show the validity of this approach by comparing inferences of adaptive Metropolis-coupled MCMC to MCMC on several datasets. We then explore where Metropolis-coupled MCMC provides benefits over MCMC. We implemented this adaptive Metropolis-coupled MCMC approach as an open source package licenced under GPL 3.0 to the Bayesian phylogenetics software BEAST 2, available from https://github.com/nicfel/CoupledMCMC.

Introduction

Phylogenetic methods are being used to study increasingly complex processes. Analyses using such methods, however, also require an increasingly large amount of computational resources. One way to still be able to perform these analyses is by making use of multiple CPU’s, which requires calculations to be able to run in parallel. Tree likelihood calculations (Suchard & Rambaut, 2009) often assume independent evolutionary processes on different branch and nucleotide sites and can be easily parallelised (Suchard & Rambaut, 2009). This can, however, be complex or even impossible for many other parts of such analyses, most notably tree prior calculations, which are used to infer demographic processes from phylogenetic trees. A lot of recent development in the field of phylogentics has been focused on developing such tree priors that allow us to infer complex population dynamics from genetic sequence data (Müller, Rasmussen & Stadler, 2018; De Maio et al., 2015), which are very computationally intensive. This is because, in contrast to tree likelihood calculations, these models often require solving equations that are dependent on each other, such as computing the location of lineages from tips to the root of trees (Müller, Rasmussen & Stadler, 2018; De Maio et al., 2015). As a result, analyses using standard Bayesian tools, such as Markov chain Monte Carlo (MCMC), can be very time consuming. This, in turn, limits the datasets that can be studied and the complexity of models that can be used to do so.

Alternatively, Metropolis-coupled MCMC (MC3) can be used to speed up analyses in Bayesian phylogenetics (Altekar et al., 2004; Ronquist et al., 2012; Aberer, Kobert & Stamatakis, 2014; Höhna et al., 2016). This approach is based on running multiple MCMC chains, each at a different ‘temperature’, which effectively flattens the posterior probability space (Geyer, 1991; Gilks & Roberts, 1996). This allows heated chains to move faster through the posterior probability space, and increases the chance to travel between local optimas (Whidden & Matsen, 2015). After some amount of iterations, two chains are randomly selected and potentially exchanged in what is essentially an MCMC move. In such a move, the parameters of the two chains are exchanged, but each chain keeps its temperatures. While the heated chains do not explore the true posterior probabilities, the one cold chain does (Geyer, 1991; Gilks & Roberts, 1996). In contrast to MCMC, however, Metropolis-coupled MCMC requires additional parameters to set up an analysis. Defining the temperatures of each chain in particular, can be problematic and may require some amount of testing. Choosing sub-optimal temperatures of chains can lead to inefficient exploration of the posterior probability space, essentially wasting the additional computational resources used (Brown & Thomson, 2018).

The problem of finding good temperatures is related to the issue of finding good variances of proposal distributions in MCMC. One way to deal with this is to automatically adapt variances in proposal distributions to achieve optimal acceptance probabilities of moves during an MCMC (Haario, Saksman & Tamminen, 2001). This can be applied to adaptively tune the temperatures of heated chains in the Metropolis-coupled MCMC framework (Miasojedow, Moulines & Vihola, 2013). We here employ this adaptive mechanism to tuning the temperature difference between chains in the Metropolis-coupled MCMC algorithm. We either use incremental heating (Altekar et al., 2004), or assume the temperature to be distributed using the quantiles of a beta distribution with α = 1 and β being a tuning parameter. The amount by which the temperature is updated is increasingly being reduced during each run, which eventually leads the temperatures of chains to be approximately constant (Haario, Saksman & Tamminen, 2001). While not being Markovian, this leads the algorithm to be ergodic.

We implemented this adaptive Metropolis-coupled MCMC algorithm in BEAST 2 (Bouckaert et al., 2014), which runs on all popular operating systems, and where a lot of novel Bayesian phylogenetic model development currently takes place (Bouckaert et al., 2019). This implementation makes use of multiple CPU cores (potentially on different computers), allowing virtually any analysis in BEAST 2 to be performed on multi-core machines or multiple machines increasing the size of datasets that can be analysed and the complexity of models that can be used to do so. By default, the implementation adapts the temperature difference between heated chains to achieve an acceptance probability of any two chains, on average, being exchanged of 0.234 (Roberts, Gelman & Gilks, 1997; Roberts & Rosenthal, 2001; Kone & Kofke, 2005; Atchadé, Roberts & Rosenthal, 2011).

We first show the correctness of the adaptive MC3 approach by comparing summary statistics of multi type tree distributions sampled under the structured coalescent (Vaughan et al., 2014) to the summary statistics received when using regular MCMC. Additionally, we show that distributions of posterior probability estimates are constant over the course of analyses using adaptive MC3, when inferring past population dynamics of Hepatitis C in Egypt (Ray et al., 2000; Pybus et al., 2003).

Next, we show how automatically tuning the temperature, leads to an acceptance probability that converges to the target probability from different initial temperatures on two different datasets.

We then compare MCMC to adaptive MC3 using different levels of heating on two different datasets. First, we apply it to the Hepatitis C dataset, where we do not expect regular MCMC to be stuck in local optimas. Then, we apply it to a dataset which has been described to be easily stuck in local optimas (Lakner et al., 2008; Höhna & Drummond, 2011).

Methods and Material

Background

Metropolis-coupled MCMC makes use of running n different chains i = 1,…,n at different temperatures (Geyer, 1991; Gilks & Roberts, 1996; Altekar et al., 2004). Each of the different chains works similar to a regular MCMC chain. In regular MCMC, a parameter space is explored as follows: Given that the MCMC is currently at state x, we propose a new state x′ from a proposal distribution g(x′|x) given the current state. At this new state, we calculate the likelihood P(D|x′) of the data D given the state and the prior probability of the new state P(x′) and compare it the to old state. The probability of accepting this new state is then calculated as follows: (1) R=min[1,P(D|x′)P(x′)P(D|x)P(x)g(x|x′)g(x′|x)]

If R is greater than a randomly drawn value between [0,1], the new state x′ is accepted as the current state, otherwise it is rejected and we remain in the same state. If we keep proposing new states x′ and accept these using Eq. (1), we eventually explore parameter space with the frequency at which values of a parameter are visited being its marginal probability (Geyer, 1991).

One of the issues of using this approach is that acceptance probabilities can be quite low, which makes it hard to move between different states in parameter space. Alternatively, an MCMC chain can be heated by using a temperature scaler βi=11+(i−1)Δt, with i being the number of the chain (Altekar et al., 2004). Heating of an MCMC chain changes its acceptance probability Rheated to: Rheated=min[1,(P(D|x′)P(x′)P(D|x)P(x))βig(x|x′)g(x′|x)]

For a heated chain, the frequency at which a value of a parameter is visited does not correspond to its marginal probability any more. However, heated chains can be used as a proposal to update the cold chain by performing what is essentially an MCMC move. This move proposes to swap the current states of two random chains i and j with the temperature βi and βj such that βi < βj. Exchanging the states of chains i and j is accepted with an acceptance probability Rij of: Rij=min[1,P(xi|D)βjP(xj|D)βiP(xi|D)βiP(xj|D)βj]

As for a regular MCMC move, swapping the states of the two chains is accepted when a randomly drawn uniformly distribution value in [0,1] is smaller than Rij.

Additional to randomly swapping states between chains, we also implemented the possibility to only swap the states of neighbouring chains. That means that we condition on i = j + 1 instead of randomly sampling both i and j.

Locally aware adaptive tuning of the temperature of heated chains

Choosing an optimal temperature of the different heated chains can be a tedious task, requiring running an analysis, updating temperatures of the analysis and re-running everything. Instead, the temperatures of chains can be tuned automatically during the run itself to achieve a targeted average acceptance probability. Ideally, we would like to adjust the temperature such that effective sample size (ESS) of parameters of interest is maximised per unit of time, but ESSs are hard to estimate while running an analysis. Therefore, optimising for average acceptance probability balances the need for moving through the MCMC’s state space (at higher acceptance probability), and making bold moves (at lower acceptance probability), which are two requirements for getting good ESSs per unit of time. As stated above, we consider the temperatures difference between the n different chains to be a constant value ∆t, which we tune during the analysis.

When updating the temperature based on the global acceptance probability, we compute pglobal based on all proposed exchanges of states from the start of a run to the current state. We then iteratively tune the temperature to achieve the target average acceptance probability ptarget over the course of an analysis as follows. Given pglobal and ptarget, we update the difference in temperature between chains ∆t as follows: (2) Δtnew=max[0,Δtcurrent+pglobal−ptarget#exchanges]

With # exchanges denoting the total number of proposed exchanges, which increases throughout the BEAST run. This means that updating the temperature as in Eq. (2), leads the tuning of the temperature to become smaller and smaller and eventually approaches zero.

Tuning ∆t is only performed after an initial burn-in period of (by default) 100 proposed exchanges. By default, the target acceptance probability is set to 0.234, which for many MCMC proposals can be shown to be an optimal trade-off between as many accepted moves as possible and as large of a move as possible (Kone & Kofke, 2005; Atchadé, Roberts & Rosenthal, 2011). Datasets where unfavourable intermediate states are of particular issue may, however, require higher temperatures and therefore lower acceptance probabilities to overcome these intermediate states.

Changing the temperature of a heated chain changes the equilibrium distribution of that chain. There can be a significant time lag between changing the temperature of a chain and that chain moving to its new equilibrium state. If the temperature is updated too fast, heated chains may not have reached this new equilibrium yet which in turn can lead to over-adaptation. This is particularly problematic at the beginning of an analysis where ∑exchanges is relatively small and where large changes in the temperature could occur. In order to reduce the risk of that, we maximise the difference between ∆tcurrent and ∆tnew, that is by how much the temperature can be changed, to be 0.001.

Another issue can arise when the global acceptance probability strongly differs from the current acceptance probability. In order to avoid that, we made the adaptation procedure aware of the local acceptance probability. To do so, we compute a local acceptance probability plocal of the last 100 proposed exchanges. We only update the temperature if the global and the local acceptance are on the same side of the target acceptance probability, that is if plocal > ptarget & pglobal > ptarget or plocal < ptarget & pglobal < ptarget.

Implementation

In this implementation of MC3, we run n different MCMC chains, with each chain i ∈ [1,…,n] running at a temperature βi=11+(i−1)Δt (Altekar et al., 2004). We additionally implemented a scenario where the values for βi are given by the quantiles of a beta distribution, such that βi=1−cdf(i−1nrchains). With cdf being the cumulative density function of a beta distribution with α = 1 and β being the tuning parameter.

Upon initialisation, we first sample at random at which iteration the states of two chains with which number are proposed to be exchanged. We then initialise each chain to be run in its own Java thread using multiple CPU cores, if available. Each chain is then run until it reaches the time when an exchange of states with another chain will be proposed. This means than every chain runs independently of each other until an iteration at which it actually participates in a proposed exchange, minimising the crosstalk between threads (Altekar et al., 2004).

This is, however, only true for swapping between random chains. When restricting swaps to only occur between neighbouring chains, we run each chain until the next possible swap. We then randomly choose between which two chains, a swapping of states is proposed.

If the exchange of states between different chains is accepted, we exchange the temperature of the two chains instead of the states themselves (Altekar et al., 2004; Ronquist et al., 2012; Aberer, Kobert & Stamatakis, 2014; Höhna et al., 2016). The states can be quite large and exchanging them across different chains is potentially quite time consuming. Instead of exchanging the states themselves, we exchange the operators and loggers, which are the objects that produce the log files. Exchanging the operator specifications is done such that the individual tuning parameters of operators of a chain can be optimised to run at specific temperatures. The loggers are exchanged such that each heated chain logs its states to the log file that corresponds to its temperature and not the number of the chain.

The temperature is adapted at any potential exchange of states between chains, after an initial phase of 100 potential exchanges without any adaption. The temperature is updated simultaneously on all chains, not just the ones participating in the exchange of states, independent of which iterations they are in.

Adaptive MC3 is implemented, such that runs that were prematurely stopped or didn’t reach sufficient convergence yet can be resumed. Usually, a graphical user interface called BEAUti is used to set up BEAST 2 analyses. Setting up analyses with MC3 works differently depending on whether a BEAUTi template is needed to set up an analysis as required for some packages. If no such template is needed, an analysis can be set up to run with MC3 directly in BEAUTi and we provide a tutorial on how to do this on https://taming-the-beast.org/tutorials/CoupledMCMC-Tutorial/(Barido-Barido-Sottani et al., 2017). Alternatively, we provide an interface that converts BEAST 2 XMLs set up to run with MCMC into such that run with adaptive MC3.

Data availability and software

The BEAST 2 package coupledMCMC can be downloaded by using the package manager in BEAUti. The source code for the software package can be found here: https://github.com/nicfel/CoupledMCMC. The XML files used for the analysis performed here can be found in https://github.com/nicfel/CoupledMCMC-Material. All plots were done using ggplot2 (Wickham, 2016) in R (R Development Core Team, 2013).

Validation

Similar to the validation of MCMC operators, we can sample under the prior to validate the implementation of the MC3 approach. To do so, we sampled typed trees with five taxa and two different states under the structured coalescent using the MultiTypeTree (Vaughan et al., 2014) package for BEAST 2. We did this sampling once using MCMC and once using MC3. If the implementation of the MC3 algorithm explores the same parameter space as MCMC, marginal parameter distributions sampled using both approaches should be equal. In Fig. S1, we compare the distribution of different summary statistics of typed trees between MCMC and MC3, which shows both methods are in agreement.

Results

Ergodicity of the adaptive Metropolis-coupled MCMC algorithm

First, we test if the distribution of posterior probability values using adaptive MC3 algorithm are consistent over time, that is ergodic. To do so, we ran 100 skyline (Drummond et al., 2005) analyses of Hepatitis C in Egypt (Ray et al., 2000), with three different target acceptance probabilities, 0.234 (Kone & Kofke, 2005; Atchadé, Roberts & Rosenthal, 2011), 0.468 (= 2 * 0.234) and 0.117 (=0.2342). The temperature difference between chains ∆t is being adapted during the analyses, particularly during the initial phase (see Fig. 1B).

Figure 1 Distribution of posterior probability values at different iterations over 100 analyses.

(A) The black line denotes the mean posterior probability estimates (y-axis) over 100 analysis at different iterations (x-axis). The grey area denotes the 95% highest posterior density interval of posterior probability estimates over these 100 analyses at different iterations. The different subplots show the results using runs with three different target acceptance probabilities, leading to different temperature differences between the chains. (B) The black line denotes the mean temperature difference ∆t between chains on the y-axis over 100 analyses at different iterations on the x-axis. The grey area denotes the 95% highest posterior density interval of ∆t over these 100 analyses at different iterations.

We then computed the distribution of posterior probability estimates of the 100 different runs using the posterior probability estimates at different iterations. The distribution of posterior probability estimates stays constant over the different iterations (see Fig. 1A), despite the temperature difference between chains being adapted. This is true for all three different target acceptance probabilities.

Automatic tuning of the temperature of heated chains

Next, we tested how well the adaptive tuning of the temperature of heated chains over the course of an analysis works starting from different initial values. To do so, we ran two different datasets, the Hepatitis C dataset (Ray et al., 2000) as well an influenza A/H3N2 analysis using MASCOT as analysed previously (Müller, Rasmussen & Stadler, 2018). We ran each dataset with four different initial temperatures (0.0001, 0.001, 0.01 and 0.1), each targeting three different acceptance probabilities, 0.234, 0.468 and 0.117. Additionally, we used two different frequencies to propose swaps between chains, once proposing swaps every 100 iterations and once every 1,000. Since the temperature is adapted at every possible swap, this means that the runs with swaps every 100 iterations adapt ∆t 10 times more frequently than the ones proposing swaps every 1,000 iterations. We kept the temperature scaler constant for the first 100 potential swaps of states between chains.

As shown in Fig. S2, for any of the here considered initial values of the temperature scaler, the target acceptance probability is reached quite early in the run and very well approximated at the end of the run using the Hepatitis C example. The same applies to the analysis of the influenza A/H3N2 dataset (see Fig. S4).

After an initial phase where the adaption of the temperature difference can overshoot the optimal value, ∆t is adapted such that it approximates the target value better and better during the run (see Fig. 2 and Fig. S3 for the MASCOT analysis).

Figure 2 Automatic tuning of the temperature to achieve different acceptance probabilities.

Here, we show how the temperature difference between chains (y-axis) is adapted during the course of an adaptive MC3 run on the x-axis. Each colour represents runs with different target acceptance probabilities. For each of the four different target acceptance probabilities, we started runs at four different initial temperatures. (A) Acceptance probability over the course of a run when swaps of states between chains are proposed every 100 iteration. (B) Acceptance probability when swaps are proposed every 1,000 iteration.

The effect of heating on exploring the posterior

In order to explore how heating affects exploring the posterior probability space, we next compared ESS values between regular MCMC and MC3 at different temperatures on a dataset where we do not expect any problems in exploring the posterior space caused by several local optimas. ESS values denote the number of effective samples if all samples would be drawn randomly from a distribution and are estimate here using Tracer (Rambaut et al., 2018).

To compare ESS values, we ran the Bayesian coalescent skyline (Drummond et al., 2005) analysis of Hepatitis C in Egypt (Ray et al., 2000) for 4 * 107 iterations using MCMC in 100 replicates. We then compare these ESS values to those received when performing the same analysis using MC3 with four different chains for 1 * 107 iterations using three different target acceptance probabilities, 0.468, 0.234 and 0.117. We also ran four times 100 additional analysis using different settings for the adaptive MC3 algorithm, all with a target acceptance probability of 0.234. First, we assume the temperature differences between chains to be distributed according to the quantiles of a beta distribution. We next allowed only swapping of states between chains with neighbouring temperature. Additionally, we estimate ESS values when running the same analysis using 8 and 16 chains for 5 * 106 respectively 2.5 * 106 iterations.

The different chain lengths between MCMC and MC3 are chosen such that the overall number of iterations over the cold and heated chains is the same for MC3 as for MCMC. After running all eight times 100 analyses, we computed the ESS values of the posterior probability estimates using loganalyser in BEAST 2 (Bouckaert et al., 2014).

As shown in Fig. 3A, the average ESS values are highest for the cold scenario when using MC3 and decrease with lower target acceptance probabilities. Lower target acceptance probabilities mean higher temperatures of heated chains in those analyses. With an increasing number of chains, but proportionally less iterations per chain, the ESS values decreases. This is particularly pronounced when using 16 chains.

Figure 3 Convergence of coupled MCMC and regular MCMC using posterior ESS values and Kolmogorov–Smirnov distances.

(A) Here, we show the distribution of effective samples size (ESS) values of the posterior probabilities after 4 * 107 iterations for regular MCMC and after 1 * 107 iterations for MC3 with 4 chains, 5 * 106 iterations for those with 8 and 2.5 * 106 iterations for those with 16 chains, so wall time for MCMC runs was much larger than for MC3. When running the analyses with MC3, we used three different target acceptance probabilities. (B) Here, we show the distribution of Kolmogorov–Smirnov distances between individual runs and the concatenation of all individual runs. We assume that all 800 runs concatenated describe the true distribution of posterior values and then take the KS distance as a measure of how good an individual run approximates that distribution. The smaller a KS value, the better the true distribution is approximated.

We next tested if higher ESS values actually correspond to a run approximating the distribution of posterior probability values better. To do so, we compared Kolmogorov–Smirnov (KS) distances between individual runs and the true distribution of posterior values. The KS distance denotes the maximal distance between two cumulative density distributions, which is smaller the better two distributions match. Since we can not directly calculate the true distribution of posterior values, we concatenated the 800 regular and MC3 runs and used the concatenated distribution of posterior values as the true distribution. While this is technically not an independent run to compare to, each individual run contributes relatively little to the reference run.

Figure 3B shows the distribution of KS distances between individual runs using regular and MC3 to what we assume to be the true distribution. In contrast to the comparison of ESS values, we find that the distribution of KS distances is fairly comparable across all methods. This indicates that in this analysis, MC3 with four individual chains performs equally well as MCMC run for four times as long. With an increasing number of chains, however, this relationship hold less and less true. While the analysis with 8 chains still leads to a similar distribution of KS values, using 16 chains leads to a higher KS values.

We additionally tested how well the true tree distribution is recovered. To do so, we computed for each individual run the posterior clade support and compared it to a reference run consisting of all 100 runs combined. We then compare the maximal difference between clade support for each individual run to the reference run and show the estimated values in Fig. S5. Overall, the same patterns as for the KS distance holds true, with the analysis with 16 chains performing the worst, while the other analyses performed comparably.

It also shows that the differences in ESS values between the MC3 runs with different target acceptance probabilities are indicative of more swaps, rather than a better approximation of the true posterior probability distribution. Using the quantiles of a beta distribution instead of incremental heating as spacing between adjecent chains did not seem to impact the ESS values nor the KS distance. Only swapping states of chains with neighbouring temperatures performs equal to randomly swapping chains, but with a lower target acceptance probability. Swaps between neighbouring chains leads to, on average, hotter chains at the same acceptance probability.

We next compared the inference of trees on a dataset (typically referred to as DS1) that has proved problematic for tree inference using MCMC (Lakner et al., 2008; Höhna & Drummond, 2011; Whidden & Matsen, 2015; Maturana Russel et al., 2018). This dataset is essentially made up of different tree islands (Whidden & Matsen, 2015). Transitioning between the different tree island is highly unlikely due to very unfavourable intermediate states, making heating necessary to travel between local optima (Höhna & Drummond, 2011; Whidden & Matsen, 2015).

We ran the dataset using MCMC for 5 * 107 iteration and MC3 for 5 * 107 with 4 different chains. We ran MC3 targeting three different acceptance probabilities, that is 0.117, 0.234 and 0.468. As shown previously (Lakner et al., 2008; Whidden & Matsen, 2015) MCMC gets stuck in different local optimas, resulting in differences between inferred clade probabilities across different runs (see Fig. 4). As above, we additionally analysed this dataset using the quantiles of a beta distribution as spacing between chains or restricted swaps to only occur between neighbouring chains. We also ran two analyses with 8 and 16 chains, but with half respectively one quarter of the iterations per individual chain, such that the overall computations remained constant.

Figure 4 Inferred clade probabilities between different replicate runs.

Here, we compare inferred clade probabilities between one run (y-axis) and four replicates from different starting points (x-axis) using MCMC (A) and adaptive MC3 run with target acceptance probabilities of 0.468 (B), 0.234 (C) and 0.117 (D). In (E), we show how well the tree space is explored when assuming the temperatures of the heated chains are distributed according to the quantiles of a beta distribution and a target acceptance probability of 0.234. In (F), we only allow swaps between neighbouring chains and in (G) and (H), we show the results when using 8, respectively 16 chains, but with only half, respectively a quarter of the iterations.

The clade probabilities are more comparable when targeting an acceptance probability of 0.468 and become more consistent between the different runs with acceptance probabilities of 0.234 and 0.117. At higher target acceptance probabilities (i.e. lower temperatures), the heating of chains is not sufficient to efficiently travel between local optimas.

We additionally compared how well the different runs approximate the posterior probability distribution compared to how long they ran. Consistent with for example Lakner et al. (2008), several MCMC runs sample from a different posterior probability distribution compared to MC3 with a low target acceptance probability and a high temperature (see Fig. S6). When running MC3 with a relatively high target acceptance probability of 0.468, the KS distance to the reference distribution decreases relatively slowly with the number of iterations compared to lower acceptance probabilities. This suggests that at lower temperatures (i.e. higher acceptance probabilities), some of the chains get stuck in local optimas (Brown & Thomson, 2018).

In all other scenarios using MC3, the KS values steadily decrease, indicating convergence (see Fig. S6). This suggest that for this dataset, the most important thing is that the temperature of at least some of the heated chains is high enough to overcome the unfavourable intermediate states. Once this is achieved there does not seem to be a big difference between the settings to explore the tree space.

Discussion

Next generation sequencing has lead to ever larger datasets of genetic sequence data being available to researcher. To study these, more and more complex models are developed, many of which are implemented in the Bayesian phylogenetic software platform BEAST 2 (Bouckaert et al., 2014). Parallelising these models can often be hard or even impossible and MCMC analyses often have to be run on single CPU cores.

Alternatively, MC3 can make use of multiple cores, but a full featured version was so far not available in BEAST 2. Parallel tempering, however, requires choosing optimal temperatures of heated chains. We here circumvent the issue of choosing optimal temperatures by adaptively tuning the temperature difference between heated chains to achieve a target acceptance probability implemented for BEAST 2.5 (Bouckaert et al., 2019). In order to only have one parameter to tune, we assume that the temperature difference between heated chains is given by a constant value ∆t, which we tune during the analysis. We show that this adaptive tuning of the temperature difference is targeting different acceptance probabilities well, starting from various different initial values. Alternatively, the temperature differences could be defined between individual chains, which would require tuning the number of chains minus 1 temperatures (Miasojedow, Moulines & Vihola, 2013). While potentially leading to a more optimal spacing of temperatures between individual heated chains, we here chose an approach where the number of parameters that have to be tuned is minimal. We hope that this minimises the amount of tuning needed and reduces the complexity of setting up an analysis to the same level as for a regular MCMC analysis and therefore makes it as user friendly as possible.

We next compared convergence between using different target acceptance probabilities, different settings of the adaptive MC3 analysis, as well as regular MCMC. We find that ESS values are comparable between MC3 with N chains and a relatively high target acceptance probability of 0.468 and regular MCMC that ran N times longer. ESS values decreased on this dataset when using lower target acceptance probabilities and therefore higher temperatures.

When comparing how well the true posterior distributions are approximated between the different target acceptance probabilities, we found that using different target values did not significantly influence how well the distributions are approximated.

ESS values are estimated by computing the auto-correlation time between samples. We suspect that swapping the states between chains strongly decreases this auto-correlation. In turn, this would mean that the more frequently states are exchanged, the shorter this auto-correlation become, which would increase ESS values. This appearance of convergence can be particularly problematic when all chains are stuck in local optimas and where swapping of states can lead (Brown & Thomson, 2018). As suggested in (Brown & Thomson, 2018), using more chains, lower acceptance probabilities (i.e. higher temperatures) and particularly, running several replicate analyses and checking convergence of heated chains can help to detect this issue. This implementation allows users to log heated chains as well, although not by default. Using additional convergence statistics like the scale reduction factor (Brooks & Gelman, 1998), might help in assessing convergence.

Since the MC3 runs required N times fewer iterations of the cold chain to approximate the distribution of posteriors values as well, MC3 can potentially help speed up analysis by a factor N that can be chosen to be proportional to the number of CPU’s used. However, this is not necessarily a linear relationship. Using 8 or 16 chains but proportionally less iterations does not lead to the same ESS values as using less chains but longer runs. This suggest that adding more chains is less and less beneficial and running several replicate analyses and then combining the runs might be a better use of computational resources. For datasets where the heating of chains is not needed to explore the posterior probability space, it might be more computationally efficient to run N independent MCMC analyses and combining them instead of running a MC3 analysis with N chains. An added benefit is that it is easier to detect convergence issues with MCMC compared to MC3. In practice, this means that using MC3 is most beneficial in cases where regular MCMC shows convergence issues, such as not being able to retrieve the same posterior distribution starting from different initial values.

The adaptive MC3 algorithm is compatible with other BEAST 2 packages and therefore works with any implemented model that does not directly affect the MCMC machinery. This will help analysing larger datasets with more complex evolutionary and phylodynamic models without requiring additional user specifications other then the number of heated chains.

Supplemental Information

Supplemental Information 1 Comparison of inference between adaptive MC3 and regular MCMC.

Comparison of the distribution of tree heights and tree lengths sampled under the structured coalescent using MultiTypeTree. The inferred distribution of tree heights and tree lengths match up between MCMC and the cold chain in MC3.

Click here for additional data file.

Supplemental Information 2 Acceptance probabilities over the course of an analyses with different target acceptance probabilities.

The global acceptance probability during the course of an adaptive parallel tempering run on the x-axis. Each colour represents runs with different target acceptance probabilities. For each of the four different target acceptance probabilities, we started runs at four different initial temperatures. A Acceptance probability over the course of a run when swaps of states between chains are proposed every 100 iteration. B Acceptance probability when swaps are proposed every 1000 iteration.

Click here for additional data file.

Supplemental Information 3 Automatic tuning of the temperature to achieve different acceptance probabilities for a MASCOT analysis of influenza A/H3N2.

How the temperature difference between chains (y-axis) is adapted during the course of an adaptive parallel tempering run on the x-axis. Each colour represents runs with different target acceptance probabilities. For each of the four different target acceptance probabilities, we started runs at four different initial temperatures. A Acceptance probability over the course of a run when swaps of states between chains are proposed every 100 iteration. B Acceptance probability when swaps are proposed every 1000 iteration.

Click here for additional data file.

Supplemental Information 4 Acceptance probabilities over the course of an analyses with different target acceptance probabilities for a MASCOT analysis of influenza A/H3N2.

The global acceptance probability during the course of an adaptive parallel tempering run on the x-axis. Each colour represents runs with different target acceptance probabilities. For each of the four different target acceptance probabilities, we started runs at four different initial temperatures. A Acceptance probability over the course of a run when swaps of states between chains are proposed every 100 iteration. B Acceptance probability when swaps are proposed every 1000 iteration.

Click here for additional data file.

Supplemental Information 5 Comparison between clade support for individual runs to reference run.

The distribution of maximal differences of clade supports for individual runs compared to a reference run. The reference run is made up of all 100 runs of an analysis.

Click here for additional data file.

Supplemental Information 6 Convergence versus number of iterations of the posterior probability distribution for different acceptance probabilities.

The Kolmogorov-Smirnov (KS) distance between the inferred posterior probability distribution and a reference posterior probability distribution on the y-axis up to the iteration on the x-axis. The different plots show the KS distance over the number of iterations for MCMC (A), and MC3 with a target acceptance probability of 0.468 B, 0.234 C and 0.117 D. In E, we assume the temperatures of the heated chains are distributed according to the quantiles of a beta distribution and a target acceptance probability of 0.234. In F, we only allow swaps between neighbouring chains and in G and H, we show the results when using 8 respectively 16 chains, but with only half respectively a quarter of the iterations. The reference distribution is made up of all MC3 runs that have a target acceptance probability of 0.234. In order to avoid comparing runs against themselves, we remove all runs with the same replicate number in the reference run for the KS calculation.

Click here for additional data file.

We would like to thank Paul Lewis, Sebastian Höhna and a third anonymous reviewer for their helpful comments on the manuscript.

Additional Information and Declarations

Competing Interests

Author Contributions

Data Availability

The authors declare that they have no competing interests.

Nicola F. Müller conceived and designed the experiments, performed the experiments, analysed the data, prepared figures and/or tables, authored or reviewed drafts of the paper, implemented software, and approved the final draft.

Remco R. Bouckaert conceived and designed the experiments, authored or reviewed drafts of the paper, implemented software, and approved the final draft.

The following information was supplied regarding data availability:

The BEAST 2 package coupledMCMC can be downloaded by using the package manager in BEAUti. The source code for the software package is available at GitHub: https://github.com/nicfel/CoupledMCMC.

The XML files used for the analysis performed here is available at GitHub: https://github.com/nicfel/CoupledMCMC-Material.

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
