# Peer review of "Adaptive Metropolis-coupled MCMC for BEAST 2"

_PeerJ, doi:10.7717/peerj.9473_

## Round 0.1 · original submission · Minor Revisions

Three specialists in the field evaluated your submission. They all see merits in your work and recommend minor revision. I agree with them.

·

Basic reporting

The paper is clearly written, cites relevant literature, provides sufficient background, and represents an appropriate unit of publication.

Some typos:

lines 100-101: "The acceptance probability of accepting this new state..." --> "The probability of accepting this new state..."

line 216: "We ran to different datasets..." --> "We ran two different datasets..."

line 256: "...which is smaller the better to distributions match.: --> "...which is smaller the better two distributions match."

Experimental design

The experimental design of this study was commendable, and clearly demonstrates that their approach to adapting the chain temperature schedule works well. There were, however, a few sections that need clarification:

a. Line 114 says "As stated above, we consider the temperatures of n different chains to be geometrically distributed..." and lines 152-153 say "The temperatures of the different chains are therefore geometrically distributed, which has been shown to be a good spacing of temperatures between individual chains...." I don't see how the temperatures can be geometrically distributed, as the geometric distribution is a discrete distribution, which would require temperatures to be non-negative integers, and, in any case, the chain temperatures are computed deterministically given delta t and are not random variables. I think the authors meant to say that the temperatures (or inverse temperatures) form a geometric progression, but even that interpretation is confusing, as their formula seems to generate an arithmetic, not a geometric, progression (i.e. there is a constant *difference* between adjacent inverse temperatures, not a common *ratio*. So, the authors need to clarify what they mean here.

b. Lines 142-148 suggest that the local and global acceptance probabilities are statistics computed from the last 100 proposed exchanges or all exchanges, respectively. That part made sense. Earlier, however, the text between lines 161-120 is a bit confusing. The statement (starting with line 117) "When updating the temperature based on the global acceptance probability, we compute p_{current} based on all proposed exchanges of states from the start of a run to the current state" seems to conflict with the statement (starting on line 119) "At each proposed exchange of states between states, we denote the probability of an exchange being accepted as p_{current}." So p_{current} is, in the first statement, based on the entire run up to that point, and, in the second statement, based only on the current iteration. How does p_{current} relate to R_{ij} (equation at the end of the paragraph lacking line numbers between line 106 and line 107)? Please clarify the relationship between p_{current}, p_{local}, and R_{ij}.

c. Do exchanges only take place between chains that have temperatures that are neighbors in the temperature progression? Or can the cold chain swap with the hottest chain when more than 2 chains are being used? If the latter, it would seem more difficult (rejection rate of proposed exchanges greater) for jumps between chains at the extremes of temperature compared to swaps between chains with adjacent temperatures. Which acceptance probability is being compared to the target acceptance probability: the average acceptance rate across all possible exchanges? Lines 74-76 seem to say that the acceptance rate between any pair of chains is identical and can be made equal to the target acceptance probability: "By default, the implementation adapts the temperature difference between heated chains to achieve an acceptance probability of any two chains being exchanged of 0.234." I don't see how the "any two chains" part of this can be correct. Please clarify whether your method instead tries to center the average acceptance rate over the target.

Validity of the findings

I thought that the analyses performed to demonstrate the effectiveness of adaptive parallel tempering were very appropriate and convincing. (Minor issue: Figure 1B is never referenced in the text as far as I can tell.) The analysis of data set DS1 (a data set that exhibits a strongly multimodal posterior) and the plots in Figure 4 are particularly compelling. I suggest that the authors also cite the following paper along with Lakner et al. 2008 and Hohna and Drummond 2011 when discussing the DS1 results. The Whidden and Matsen paper does a nice job of showing the bimodality of DS1's tree space visually, and also points out that MCMCMC is necessary for this data set to achieve adequate mixing.

C Whidden, FA Matsen IV 2015 Quantifying MCMC exploration of phylogenetic tree space. Systematic biology 64 (3), 472-491.

Additional comments

I think this is a valuable contribution to the Bayesian phylogenetics literature. The authors did not invent adaptive parallel tempering, but they demonstrate that it works well for phylogenetic applications. They clearly describe their method so that it would be easy to implement in other software, and provide an open source software solution in the form of a plugin for the widely used Bayesian phylogenetics software package BEAST2.

·

Basic reporting

The manuscript "Adaptive Parallel Tempering for BEAST 2" describes a method of
incorporating MC^3 in BEAST to hit target acceptance rates. Overall, this will be very useful and the authors show advantages of their implementation. The text should be improved both for clarity and language editing. Additionally, the current manuscript only provides some very basic testing and does not really explore different possible implementation (suggested below). For this manuscript to have more value than a description of a software implementation, namely a discussion and evaluation of method, it needs more comparisons to alternative method choices.


The manuscript never mentions MCMCMC or MC^3, although the name of the package is called "CoupledMCMC" and not "ParallelTempering". In order to reach the target
audience (phylogenetics users and methods developers), the authors should instead use the term Metropolis-Coupled MCMC, MCMCMC, or MC^3 in the manuscript, and change the working title to "Adaptive MC^3". This would also avoid confusion about differences to the MC^3 method implemented in MrBayes.

The manuscript lacks citations to support claims made throughout the manuscript. Specific examples are:
- the introduction needs examples and citations (line 36-43, line 52-56)
- the implementation follows other software , e.g., MrBayes and ExaBayes, and should be cited accordingly (e.g., swapping of temperatures only is common practice, lines 155-168)
- why is parallelising these models hard? Examples? Citations? (line 292-296)

Several sections/paragraphs/sentences were not clear or difficult to understand (only those very familiar with the subject may be able to follow). Specific examples are:
- lines 116 - 122 (especially the p_current, and p_target, which are not properly defined)
- lines 167-169 (what are loggers?).
- what are the ESS compute for in Figure 3?


Overall, the manuscript needs more polishing regarding the language and grammar. Specific example are:
- "and the tune the" (line 115)
- "to"-> two (line 216)
- iterations (line 273)
- add "iterations" after 10^7 (line 274)

More comments:
- what do the authors mean by two chains are "randomly" exchanged (line 49)?
- does the implementation work only on a single hardware? Please clarify and be more specific (line 71/72).
- please use clade probabilities instead of clade credibilities.
- In equation (2), I found it easier to understand if the \sum exchanges is replaced by #exchanges.

Experimental design

The performed tests are bit short and should be more comprehensive. Specifically, the authors should:

1) test another temperature scheme, for example, a discretized beta distribution. When using the quantiles of the beta distribution, as commonly done for "stepping stone sampling", you could adapt one or two parameters of this distribution similar to the geometric distribution.

2) Add more empirical datasets especially showing a Figure like Figure 4 (clade probabilities). Is this improvement seen also for "easier" datasets because the authors show only the continuous parameters but not the trees for their other example?

3) The authors stick to only 4 chains within an analyses. For standard HPC settings, we have 8-80 CPUs per node. Could the authors also test the impact of adding more heated chains, e.g., using 4, 8, 16, 32 and 64?

4) The authors only explore one type of chain swapping; randomly picking two chains. Often, it is more efficient if neighboring chains are swapped. Could the authors add a comparison between the random swap and neighbor swap, and a combination of both.

5) Could the authors explore different update methods and justify their choice? For standard MCMC methods, the proposal are commonly auto-tuned. Thus, there is a considerable literature on how the auto-tuning can be done.

Validity of the findings

Regarding Figure 4, how sure can we be that the reference run (x-axis) actually sampled the correct posterior probabilities? I suggest to follow the standard practice and compute some "golden runs" (see Höhna and Drummond 2012, or Whidden and Matsen 2017; Quantifying MCMC Exploration of Phylogenetic Tree Space).

The implementation of MC^3 in BEAST 2 is an interesting subject; however, the
concept is not novel. Consider comparing it to other existing implementations
(e.g., MrBayes, ExaBayes & RevBayes), and describe what the improvements are.

Additional comments

There was a recent paper by Brown and Thompson (https://academic.oup.com/sysbio/article/67/4/729/4866058) that discusses a problematic scenario for MCMCMC. Could the authors add at least one paragraph discussing how their method would behave in the given problem case/

Please provide some guidelines for users. Should users always run an MCMCMC analysis with XXX chains or first try a standard MCMC run and then use MCMCMC if the MCMC failed?

Was there smoothing performed for the posterior mean in Figure 1? If so, then please state.

The paper gives the impression that adding more chains comes for free (lines 326-331) as it states that standard MCMC uses N times the number of iterations. However, please state fair and clearly that all approaches used here take the same overall number of computing operations. If it is suggested that multi-core computers have unused CPUs, then several independent MCMCs could be ran as well and combined at the end.

Reviewer 3 ·

Basic reporting

no serious issue. See Comments to Authors

Experimental design

no concerns

Validity of the findings

no concerns

Additional comments

This is a nice contribution that I think should be accepted for publication. I have only 1 suggestion that could arguably be called "major."

major:
I think the authors should discuss how their method differs from the tuneHeat option in mcmcmc in RevBayes.

minor:

1. I think parallelisable should be spelled parallelizable.

2. line36-43. I agree that a lot of work is focussed on better tree priors, and that some aspcets of those are hard to parallelize. But I find it a bit hard to believe that these calculations are the rate-limiting step in many (any?) real analyses given the computational burden of the pruning algorithm. Is that what the authors are saying? Are they talking about things like the priors on gene trees in the Multi-lineage structured coalescent? This is a bit vague. More precision or references would help.

3. Use either "BEAST2" or "BEAST 2" consisitently.

4. I certainly wouldn't suggest this as a mandatory change, as there are no strict rules in statistical notation, but... x is very commonly used for the data in expressions of Bayes rule, and there is a general tradition of using greek letters for statisical parameters. Substituting \theta for x would help readablility by following that tradition.

5. line 125. It looks like it approaches 0. It does not "become 0" unless you mean "rounds to 0"

6. line 127. I certainly don't object to targetting the acceptance probability as a practical way to improve efficiency. It seems like the ms should discuss that in an ideal world, we'd optimize ESS or PSRF per unit of CPU time, but those are harder to calculate on-the-fly. So, targetting the acceptance probability is a good solution. It seems implausible to me that we know the correct acceptance probability to target down to the thousandths place, but that is not a big concern.

7. line 140 : "In order to reduce the risk of that, we maximize the difference
between ∆tcurrent and ∆tnew to be 0.001."
does that mean you that eqn (2) should have a min(0.001, \frac{p_current - p_target}{\sum exchanges}) ?
if so, just correct the eqn (my rough interpretation probably needs an absolute value for the min check). If not correct the text.

8. MultiTypeTree is such a vague name, that you should probably point out on line 193 what it is (even though you make the connection to "multi type trees" and the structured coalescent earlier.)

9. I think that Kolmogorov–Smirnov is usually hypenated. (it is not in figure 3 caption, though)

---

## Round 0.2 · accepted · Accept

I think you have addressed all minor comments raised by the reviewers and now the manuscript is ready for publication.